# Difficult Cases of Paroxysmal Nocturnal Hemoglobinuria: Diagnosis and Therapeutic Novelties

**DOI:** 10.3390/jcm10050948

**Published:** 2021-03-01

**Authors:** Bruno Fattizzo, Fabio Serpenti, Juri Alessandro Giannotta, Wilma Barcellini

**Affiliations:** 1Fondazione IRCCS Ca’ Granda Ospedale Maggiore Policlinico, University of Milan, 20122 Milan, Italy; fabio.serpenti@unimi.it (F.S.); jurigiann@gmail.com (J.A.G.); wilma.barcellini@policlinico.mi.it (W.B.); 2Department of Oncology and Oncohematology, University of Milan, 20122 Milan, Italy

**Keywords:** paroxysmal nocturnal hemoglobinuria, eculizumab, complement inhibitors, bone marrow failures, myelodysplastic syndromes

## Abstract

Paroxysmal nocturnal hemoglobinuria (PNH) is an intriguing disease that can pose many difficulties to physicians, as well as to hematologists, who are unfamiliar with it. Research regarding its pathophysiologic, diagnostic, and therapeutic aspects is still ongoing. In the last ten years, new flow cytometry techniques with high sensitivity enabled us to detect PNH clones as small as <1% of a patient’s hematopoiesis, resulting in increasing incidence but more difficult data interpretation. Particularly, the clinical significance of small PNH clones in patients with bone marrow failures, including aplastic anemia and myelodysplastic syndromes, as well as in uncommon associations, such as myeloproliferative disorders, is still largely unknown. Besides current treatment with the anti-C5 eculizumab, which reduced PNH-related morbidity and mortality, new complement inhibitors will likely fulfill unmet clinical needs in terms of patients’ quality of life and better response rates (i.e., responses in subjects with C5 polymorphisms; reduction of extravascular hemolysis and breakthrough hemolysis episodes). Still, unanswered questions remain for these agents regarding their use in mono- or combination therapy, when to treat, and which drug is the best for which patient. Lastly, long-term safety needs to be assessed in real-life studies. In this review, we describe some clinical vignettes illustrating practical aspects of PNH diagnosis and management; moreover, we discuss recent advances in PNH diagnostic and therapeutic approaches.

## 1. Introduction

Paroxysmal nocturnal hemoglobinuria (PNH) is an acquired hematologic disorder caused by somatic mutations of the PIGA gene in hematopoietic stem cells [1]. Such mutations result in impaired production of glycosyl phosphatidyl inositol (GPI), an anchor molecule for many different cellular membrane proteins. The latter include CD55 (also named decay accelerating factor, DAF) and CD59 (membrane inhibitor of reactive lysis, MIRL), which are natural complement inhibitors and are lost on the membrane of PNH cells. Their absence leads to suboptimal complement inhibition and complement-mediated hemolysis of red blood cells (RBCs) [1]. In the last ten years, progress has been made concerning PNH in regards to both diagnosis and therapeutic approaches: high sensitivity flow cytometry enabled us to detect very small PNH clones [2], and the complement inhibitor eculizumab (ECU), a monoclonal antibody targeting C5, has significantly reduced hemolysis, anemia, and occurrence of thrombosis, reducing PNH-related morbidity and mortality [3]. In the clinical practice, many difficulties may be encountered at diagnosis, since many confounders may be present and clinical suspicion is fundamental to initiating straightforward flow-cytometry-based diagnosis; moreover, it may be difficult to decide when to start therapy with ECU for the individual patient, and responses are suboptimal in a significant proportion of cases. In this review, we describe old and new clinical and therapeutic aspects of PNH through the presentation of some instructive clinical vignettes.

## 2. Pathogenesis

Chronic intravascular hemolysis is the main feature of PNH. It is mediated by chronic activation of the alternative pathway of complement (APC), which is part of the innate immune system. The latter is persistently activated in order to ensure constant surveillance. This pathway consists mainly of two steps: C3 and C5 convertase amplification (via a protease cascade that includes factors B and D), and membrane attack complex (MAC) formation on the targeted antigen (Figure 1A) [4]. Normal RBCs are protected from complement activations by specific membrane proteins, namely CD55, which stabilizes C3 and C5 convertase, and CD59, which inhibits MAC formation. Their deficiency leads to complement activation on RBCs and to subsequent intravascular hemolysis typical of PNH [1]. The proportion of the hematopoietic cells bearing these alterations (especially the granulocytic and monocytic cell fractions) represents the size of the PNH clone. The expression of CD55 and CD59 on patients’ cells can be normal (type I cells), reduced (type II), or completely absent (type III), correlating with different severities of hemolysis [1,5]. This phenotypic mosaicism is also explained by the possible presence of different PIGA mutations coexisting in the same patient [6].

The classic pathway is initiated by immune complexes that interact with C1q, C1r, and C1s, which act on C2 and C4, leading to the formation of C4bC2a complex (C3 convertase). The lectin pathway is activated by mannose-binding lectins found on the surface of pathogens, and generates the same C3 convertase. The alternate pathway is spontaneously activated at a low rate by a mechanism called tickover of C3, via the activation of factor B and D and stabilization by the plasma protein properdin. This mechanism also generates a C3 convertase, with a powerful amplification loop. C3b initiates the terminal complement cascade by the formation of the C5 convertase, which cleaves C5 and thereafter C6, C7, C8, and C9. This results in the formation of the terminal membrane attack complex (MAC), which forms pores in the membrane of erythrocytes promoting cell lysis (intravascular hemolysis). During complement activation, several products are powerful anaphylatoxins capable of inducing chemotaxis, cell activation, inflammation, and extravascular hemolysis by immune effectors. Thrombotic events and a variable degree of bone marrow failure occur in an unpredictable proportion of patients and may also change over time. AA: aplastic anemia; MDS: myelodysplastic syndrome; DVT: deep venous thrombosis [1,7].

## 3. Clinical Presentation and Diagnosis

### 3.1. Clinical Vignette 1 

An 18-year-old girl presented to the emergency department with disabling headache and shortness of breath. Medical history was negative except for asthma during childhood, and the patient had started a birth control pill three months earlier. Physical examination showed pale skin, scleral jaundice, and tachycardia. Brain CT scan evidenced a thrombosis of the transversal sinus, and the patient was anticoagulated with heparin. Laboratory tests showed increased D-dimer, normal coagulation times and fibrinogen, and severe hemolytic anemia (Hb 7.8 g/dL and LDH 7 × upper limit of normal, ULN), normal platelets (PLTs) and white blood cells, and increased reticulocytes (220 × 10^9^/L). Direct antiglobulin test (DAT, also known as Coombs test) was negative, and the hematologist suggested a flow cytometry assay that showed a PNH clone of 88% on granulocytes, 90% on monocytes, and 38% on erythrocytes. A diagnosis of classic hemolytic PNH was established, and the patient was transferred to a tertiary center for ECU treatment.

The clinical picture of PNH is usually dominated by signs and symptoms of the intravascular hemolysis, with anemia, jaundice, and dark urine due to the loss of free hemoglobin (Hb) from blood to urine (Figure 1B). The name “paroxysmal nocturnal” derives from the assumption that hemolysis worsens overnight due to the physiologic acidosis that occurs during sleep; it was subsequently proven that hemolysis in PNH patients is chronic and occurs throughout the day [1]. Due to intravascular hemolysis, free Hb firstly binds and saturates haptoglobin, then it irreversibly binds to nitric oxide (NO), depleting its stores. This results in vasal constriction/vasospasms, with consequent PNH symptoms including abdominal pain, dysphagia, erectile dysfunction, bone pain, and headache. Chronic uncontrolled hemolysis may also lead to progressive renal damage due to iron and heme accumulation in tubules and obstruction due to pigmented cylinders. Moreover, chronic vasospasms may lead to pulmonary hypertension [8]. PNH is also dominated by increased thrombotic risk, which may be the first presentation sign, and is reported in as many as 40% of patients [7]. Thrombosis tends to be more common in the venous district and may also occur in atypical sites, including the cerebral, mesenteric, and renal districts. Hepatic veins may also be involved and configure the so-called Budd–Chiari syndrome. Not all PNH patients experience thrombosis, and many adjunctive congenital (e.g., factor V Leiden or factor II mutations, protein C and S deficiencies, etc.) or acquired risk factors (e.g., cigarette smoke, diabetes, obesity, estroprogestin compounds, pregnancy, surgery, prolonged immobilization, etc.) may be implied, and should be looked for, as in the clinical vignette. Moreover, the degree of hemolysis is often proportional to PNH clone size, as some studies have suggested that larger PNH clones correlate with increased thrombotic risk [9].

### 3.2. Clinical Vignette 2

A 62-year-old woman presented to an internal medicine specialist due to persistent fatigue, abdominal and bone pains, and a long history of chronic macrocytic anemia. The patient was obese, suffering from fibromyalgia, and had been seen by several clinicians in the past 10 years and been diagnosed with anemia of chronic inflammation. She had also received a course of steroids with the hypothesis of suffering from DAT-negative autoimmune hemolytic anemia without response. Medical history was also positive for a superficial thrombophlebitis of the left basilic vein and mild chronic kidney disease (glomerular filtration rate 50 mL/min). During the visit, the patient mentioned recurrent urinary tract infections, which, upon deeper investigation, turned out to be dark urine episodes during upper respiratory infections. Laboratory tests showed normocytic anemia with increased red cell distribution width, consistent with the presence of increased reticulocytes and iron deficiency (ferritin 8 ng/mL). LDH was 3× ULN, unconjugated bilirubin increased, and DAT confirmed negative. Flow cytometry showed the presence of a 38% PNH clone size on granulocytes, 43% on monocytes, and 30% on RBCs. A diagnosis of classic hemolytic PNH was established.

This case shows how PNH clinical presentation may be smoldering and insidious so that clinical suspicion and careful anamnesis are fundamental. PNH diagnostic algorithms should include whole blood count, hemolytic markers (LDH, haptoglobin, absolute reticulocyte counts, and bilirubin), urine analysis for hemosiderin in urine, iron status tests, vitamin B12 and folate status, and hepatic and renal function tests. DAT is usually negative, although two cases of PNH clones in autoimmune hemolytic anemia (AIHA) have been described [10]. Other rarer causes of hemolytic anemia, such as congenital forms (hereditary RBC membrane or enzyme defects), are more typically suspected in young patients with positive family history, although some mild defects may become evident only in adulthood or when comorbidities develop. Moreover, other acquired hemolytic forms to be considered include mechanic causes (e.g., intravascular devices) and microangiopathies (e.g., disseminated intravascular coagulation thrombotic thrombocytopenic purpura, and hemolytic uremic syndrome), which may be suggested by careful medical history, by the presence of schistocytes in the blood smear, and the alteration of coagulation and/or platelets values. Bone marrow examination with morphologic, immunophenotypic, cytogenetic, and histologic evaluation is suggested, especially in patients with multiple cytopenias, to correctly diagnose a bone marrow failure (BMF) syndrome, including aplastic anemia and myelodysplastic syndromes. Once the diagnosis of DAT-negative intravascular hemolytic anemia is made, PNH diagnosis is established through a flow cytometry. The latter demonstrates the absence of at least two GPI-anchored proteins on blood cells (e.g., CD55/59). Other less sensitive tests (e.g., Ham test, the complement-lysis sensitivity test, and the sucrose test) may be used if flow cytometry is not available/accessible. Importantly, clone size should be evaluated on granulocytes/monocytes since hemolyzed RBC and transfusions may lead to underestimation of PNH clone [2].

### 3.3. Clinical Vignette 3

A 43-year-old man presented to the hematologist with pancytopenia requiring RBC and platelet transfusions. Reticulocytopenia was also noted, and bone marrow evaluation showed severe hypoplasia consistent with the diagnosis of severe aplastic anemia (infections screening negative, normal nutrients, kidney, liver, and thyroid functions). Increased LDH levels were noted, DAT test was negative, and flow cytometry revealed a PNH clone of 20% on granulocytes. The patient was admitted to hospital and treated with steroids, rabbit anti-thymocyte globulin, and cyclosporin A, and progressively recovered. Two years later, PNH clone had increased to 68% and the patient had moderate anemia, recurrent dark urine, and fatigue.

The International PNH Interest Group (IPIG) proposed a clinical classification, identifying classic hemolytic PNH, PNH associated with BMF syndromes, and subclinical forms (Figure 2) [11]. Classic hemolytic PNH is usually dominated by hemolytic anemia, PNH symptoms, and thrombosis, and is usually characterized by larger PNH clones (i.e., >50%), as in the clinical vignette 1 and 2. PNH in the context of a BMF is clinically marked by leucopenia and thrombocytopenia, with infectious/bleeding tendency, and by ineffective reticulocyte compensation, as in the clinical vignette 3. Reticulocytopenia should be carefully evaluated, by also excluding nutrients deficiencies and iron loss due to the chronic hemosiderinuria that may impair bone marrow compensation. Small PNH clones are reported in 20% to 60% of BMF cases [2,12], depending on the sensitivity of the flow cytometric technique. In fact, whilst standard flow cytometry has a sensitivity of 1%, and most laboratories report as positive only PNH clones > 10%, novel highly sensitive tests can detect clones as small as 0.01% of a patient’s hematopoiesis. Among these tests, the use of the fluorescently labeled bacterial toxin “aerolysin” (FLAER), which binds GPI with high affinity, is the gold standard [2]. The detection of even small and very small clones has been associated with better response to immunosuppression and favorable outcome in BMF, and current guidelines suggest PNH testing in these patients at diagnosis and during follow-up [9]. Pathogenically, PNH clones may represent the “residual hematopoiesis” spared by the immune attack against bone marrow precursors typical of BMF. Consistently, PNH clone size may increase after immunosuppression and recovery in these patients, as observed in clinical vignette 3. In some cases, anti-complement therapy may be required, either concomitantly (i.e., in the presence of active hemolysis and thrombosis) or even years after immunosuppression. Finally, subclinical PNH is described as the presence of clones < 10% in the absence of clinical and laboratory evidence of hemolysis [11]. The availability of FLAER has increased the detection of small PNH clones even in conditions not commonly associated with PNH, such as myeloid neoplasms, idiopathic cytopenia/dysplasia of unknown significance (ICUS/IDUS), hypomegakaryocytic thrombocytopenia, and in the general population [10]. The question arises whether these small clones are clinically or prognostically relevant, and only ad hoc future studies will clarify this issue, although it is generally recommended to distinguish PNH as a “disease” (combination of clinical symptoms, hemolysis, and identifiable PNH clone) from PNH clones as a “laboratory finding”. Table 1 summarizes the clinical and laboratory features of PNH.

## 4. PNH Therapy

In the last decade, ECU, a monoclonal antibody against C5 complement fraction, has revolutionized PNH natural history, reducing the incidence of anemia, transfusion dependency, and thrombosis, and thus increasing patients’ quality of life. Mortality and morbidity have been dramatically reduced, with a life expectancy close to that of the general population [3]. Importantly, ECU can be safely administered and is also effective in pregnant women. After a loading dose (four weekly infusions, 600 mg each), the standard dose is 900 mg intravenously (IV) every 14 days. Main indications for starting treatment are transfusion-dependent anemia, PNH clone > 10% with at least one sign/symptom related to PNH, thrombosis, or pregnancy [13]. In the clinical vignettes, patient 1 had a clear indication to start ECU as soon as possible, given severe anemia and thrombosis. In patient 2, it may be argued that mild anemia and symptoms may allow a reasonable quality of life without the high burden of the medicalization of fortnightly infusions. As a matter of fact, ECU does not eliminate PNH clone and thus must be administered lifelong [13]. Many grey zones exist regarding treatment indication, particularly in regard patients with BMF, where clone size may be difficult to assess (e.g., cases involving recent transfusions, extremely low granulocytes) and anemia is multifactorial.

### 4.1. Clinical Vignette 4

A 25-year-old female patient followed for classic hemolytic PNH, characterized by mild anemia, episodical abdominal pain, no transfusions or thrombosis, became pregnant. After a long discussion with the hematologist and the gynecologist, ECU was started and continued throughout pregnancy, post-partum, and puerperium, with no complications, and then stopped.

An exception to lifelong ECU treatment is pregnancy, which confers a transitory increase in thrombotic risk that may be controlled by ECU treatment until the end of puerperium. Before ECU, the management of a pregnant woman with PNH was characterized by increased rate of premature labor, fetal loss, and high incidence of thrombotic complications. The use of ECU in this setting was shown to be safe, to allow a favorable outcome both for the mother and the newborn, and to not interfere with breast feeding [14]. In general, besides clear indications such as transfusion dependent anemia and thrombosis, the particulars of anti-complement treatment should be always discussed with the patient, evaluating expected benefits and possible risks (discussed later).

### 4.2. Clinical Vignette 5

A 56-year-old man received the diagnosis of hemolytic PNH during admission to an internal medicine department for pneumonia and acute severe anemia. Transfusion dependency and marked hemolysis persisted despite infection recovery. After hematologic referral, the patient was vaccinated against *Neisseria meningitidis* (A, C, Y, W135, and B serogroups) and put on ECU treatment. After 6 months of treatment, LDH had completely normalized, but Hb was still around 8 g/dL, with bilirubin and reticulocytes increase and haptoglobin consumption. DAT proved positive for C3d, and the patient received 1 mg/kg/day steroids at the local hospital with only mild increase of Hb levels, occasional transfusions, and several side effects.

There are various causes of suboptimal response to ECU [15]: firstly, the drug does not work in all subjects, as in rare carriers of C5 polymorphisms [16]; moreover, some patients experience the so-called “breakthrough hemolysis” (BTH), which is the reactivation of intravascular hemolysis with LDH elevation and dark urine [17]. BTH may occur before the next ECU administration due to reappearance of complement activation (pharmacokinetic BTH), or at any time due to triggering agents (e.g., infections, surgery, trauma (pharmacodynamic BTH)). The first may be managed by increasing the ECU dose or shortening administration intervals, whilst the second may require additional doses, particularly for acute cases. Another cause of suboptimal response to ECU is that complement cascade upstream to C5 is still active and leads to C3 deposition on RBCs with consequent chronic extravascular hemolysis. This is clearly demonstrated by DAT positivity for C3d in these patients without the presence of a true autoimmune hemolytic anemia [18,19]. In these cases, steroid treatment is not indicated since it is usually ineffective, as in the clinical vignette. Splenectomy and selective splenic vein embolization have been anecdotally reported with uncertain efficacy. Additionally, ECU is not effective for the possibly associated BMF, and reticulocytopenic patients should receive bone marrow evaluation and be considered for immunosuppressive treatment or recombinant erythropoietin stimulation if aplastic anemia or myelodysplastic syndrome is diagnosed [13]. Substantial efforts have been undertaken in the last decade to innovate PNH treatment in order to fulfill these unmet needs and increase patients’ convenience.

### 4.3. Clinical Vignette 5 (Second Part)

After about 3 months of steroids, the patient had developed diabetes and hypertension, was tapered off, and finally referred to a tertiary hematologic center for PNH. All causes of suboptimal response to ECU were analyzed, and C3 mediated extravascular hemolysis was evaluated as the principal cause. The patient was further vaccinated against *Hemophilus influenzae* and *Streptococcus pneumoniae* and enrolled in a clinical trial with an oral inhibitor of the alternative pathway in addition to ECU. After a few weeks, his Hb levels progressively normalized with transfusion independency.

A plethora of novel complement inhibitors are under active study (Table 2). The new approaches aim at increasing the drug half-life of C5 inhibitors like ECU, at producing small molecules that could be administered orally or subcutaneously (sc), and at targeting the complement cascade upstream to C5 (including C1, C3, and the alternative pathway factors B and D) [20,21]. Even more recent technologies include the use of small interfering RNA (siRNA) molecules capable of inhibiting C5 synthesis, and avant-gardist gene therapy [22]. Preclinical and clinical studies have demonstrated that novel C5 inhibitors (ECU biosimilars, ECU with longer half-life such as ravulizumab IV or sc [23], small molecules [24,25], and siRNA [26]) effectively inhibit complement-mediated hemolysis and may imply more manageable administration routes/schedules. In particular, ravulizumab has a longer half-life, and was shown to not be inferior to ECU and to reduce the incidence of BTH. Additionally, some of the new agents also proved effective in patients with C5 polymorphisms [21]. Importantly, proximal inhibitors (anti-C1 iv, anti-C3 sc, and oral anti-factor B and D) were effective both in treatment of naïve patients and in those with suboptimal response to ECU, with an evident effect also on extravascular hemolysis. These agents are being studied in combination with ECU but also as single agents, with promising results [21]. However, residual C5 activation may still occur despite upstream complement inhibition, and Mannes et al. recently identified a “C3 bypass” model of terminal complement activation. This happens because of a conformational change in C5, which adopts a C5b-like structure on highly opsonized surfaces, allowing the formation of MAC complex. These findings may explain residual complement activation upon both C5, C3, and C1 inhibitors [27,28]. Finally, it is worth reminding that the only curative treatment of PNH is allogeneic bone marrow transplant (BMT). BMT outcome has been shown to significantly improve after the introduction of reduced intensity conditioning, and the addition of ECU has been shown to reduce the morbidity and mortality linked to PNH [29]. On the whole, BMT is indicated in patients with PNH and severe aplastic anemia below 40 years of age and with available donor, and has the potential to eradicate both aplastic anemia and the PNH clone. Also, PNH-MDS patients may benefit from BMT if indicated for MDS itself (e.g., high-risk patients or low-risk MDS with life-threatening cytopenias and/or molecular abnormalities).

## 5. Management of Infectious and Thrombotic Risks

### 5.1. Clinical Vignette 6

A 22-year-old boy suffering from hemolytic PNH under ECU treatment presented with fever and dark urine four days after the last ECU infusion. Laboratory tests showed the presence of moderate BTH (Hb 9 g/dL with LDH 3× ULN), and IV ceftriaxone was promptly started together with supportive treatment. Blood cultures were positive for Gram-negative diplococci sensitive to cephalosporins. After four days of antibiotics, fever resolved and C reactive protein decreased to levels close to normal, but BTH worsened and a supplementary dose of ECU was administered with progressive response. Serogroup analysis unveiled the presence of a meningococcus C sepsis for which the patient had been vaccinated.

PNH can be complicated by infectious events, either related to the coexistence of BMF and immunosuppression, or to complement inhibition. Anti-*Neisseria meningitidis* vaccinations (serogroups A, C, Y, W135, and B) are mandatory before administering ECU; if emergency treatment is started, antibiotic prophylaxis with drugs active against *Neisseria meningitidis* is warranted [30]. Novel agents, although displaying a good safety profile in clinical trials, still lack long-term data, and anti-*Streptococcus pneumoniae* and anti-*Hemophilus influenzae* vaccines are also required before enrollment. Still, infectious risk remains high for these patients since not all vaccinated subjects will get proper immunization (as in the clinical vignette) and antibody titers are not routinely evaluated. Therefore, careful patient education is crucial to recognize signs and symptoms of infections, to promptly start antibiotic therapy, and to refer to medical attention. As already discussed, infections may trigger pharmacodynamic BTH, and an additional dose of ECU may be required.

### 5.2. Clinical Vignette 7

A 56-year-old man was diagnosed with classic hemolytic PNH during hospital admission for segmental pulmonary embolism. Thrombophilia screening was negative for concomitant congenital or acquired causes. The patient started anticoagulation with heparin, then switched to oral warfarin, and ECU treatment was instituted. One year after, CT scan showed thrombosis resolution and echocardiography normal pulmonary pressures. The patient was continued on lifelong oral warfarin.

As reported above, an important issue regarding PNH is the thrombotic risk and the subsequent need for anticoagulation [8]. The possibility of stopping anticoagulation once proper anti-complement therapy is instituted in patients without concomitant risk factors for thrombosis is still an open issue, but, as of today, guidelines still recommend lifelong anticoagulation, as in the clinical vignette [31]. Another point of consideration is the possibility to use novel direct oral anticoagulants (DOACs), which are still not approved for PNH-related thrombosis. Only prospective studies will answer these questions, although some subjects are already receiving DOACs in clinical practice [32]. Finally, anticoagulant prophylaxis should be discussed for PNH patients not receiving anti-complement therapy, particularly if active hemolysis and large PNH clones are present. Referral to an expert in thrombosis and thrombophilia screening is suggested in these patients.

## 6. Conclusions

PNH is a rare, heterogeneous, and potentially fatal disease, also defined as “the great impersonator”. The disease has to be recognized quickly, as its morbidity and mortality can be dramatically reduced with early and proper intervention (monitoring, anticoagulation, complement inhibitors) and with patient education. In the clinical vignettes, we showed this heterogeneity by describing an acute and insidious presentation, disease associations (AA and MDS), and some special settings, including thrombosis, infections, and pregnancy. Moreover, we discussed who, when, and how to treat PNH, as well as the new options for suboptimal responders to standard complement inhibitor. On the whole, the advice for the clinician is to keep high suspicion and to perform PNH testing in all hemolytic conditions, in patients with BMF, and thrombosis in atypical sites. It is important to distinguish florid PNH “disease” from small PNH “clones”, as the latter can be detected in many conditions by new sensitive techniques and their significance is still debated. Initiating treatment has to be weighed on disease activity and patient’s characteristics, taking into account his/her convenience and quality of life. The availability of new agents will likely improve PNH management in the near future, fulfilling several unmet needs, including suboptimal response to ECU and BTH. Finally, patient education about the risk of infections and thrombosis is crucial, and these entities should not be disregarded even after correct prophylaxis has been established.

## Figures and Tables

**Figure 1 jcm-10-00948-f001:**
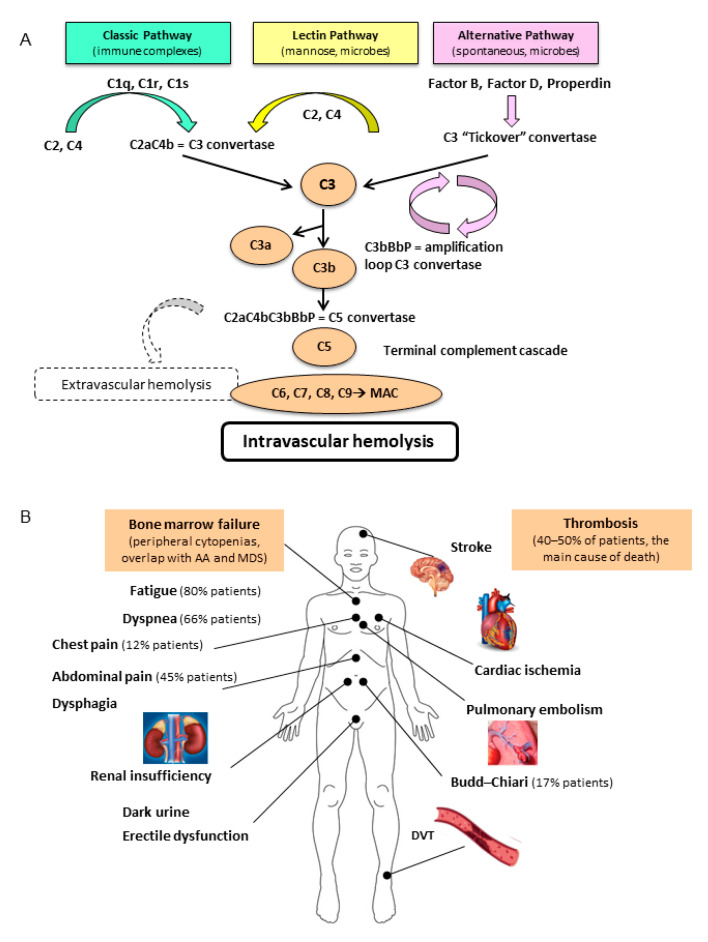
(**A**) The classic, lectin, and alternate pathway of complement activation; (**B**) The heterogeneous clinical features of paroxysmal nocturnal hemoglobinuria (PNH).

**Figure 2 jcm-10-00948-f002:**
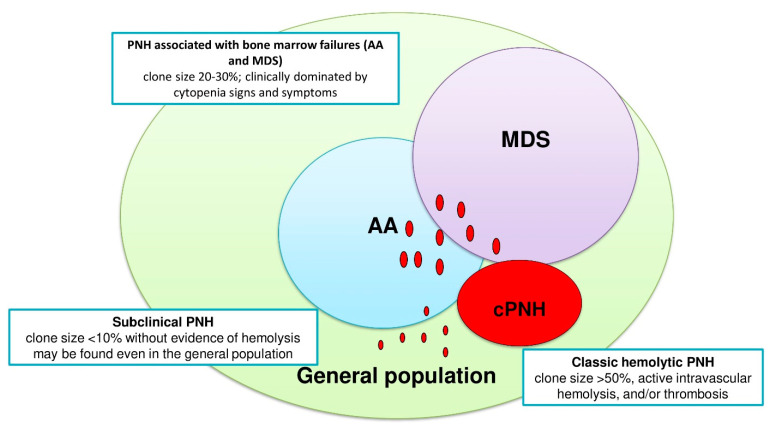
Paroxysmal nocturnal hemoglobinuria (PNH) subgroups according to the International PNH Interest Group (IPIG) classification: classic hemolytic PNH (cPNH), PNH associated with bone marrow failure syndromes, such as aplastic anemia (AA) or myelodysplastic syndrome (MDS), and subclinical PNH (sPNH). Red circles indicate PNH clones of various sizes: maximal (usually >50%) in classic PNH, intermediate (20% to 30%) in AA and MDS, and small or very small (<10% and <1%) in subclinical PNH.

**Table 1 jcm-10-00948-t001:** Clinical and laboratory characteristics of PNH.

TEST	RESULTS IN PNH	COMMENTS
Hb	↓↓↓	Anemia from mild to very severe, usually macrocytic normocromic
MCV	↓ to ↑↑	If reduced, consider coexisting iron deficiency
Reticulocyte	↓ to ↑↑	If reduced, consider coexisting nutrients/iron deficiencies or associated BMF
LDH	↑↑↑	Consider possible confounders (liver/tissue damage, folate/B12 deficiency)
Haptoglobin	↓↓↓	Possibly reduced in case of liver insufficiency or hereditary hypohaptoglobinemia
Bilirubin	↑	Usually unconjugated; conjugated bilirubin may increase in Budd–Chiari syndrome
PLT	= to ↓	If reduced, consider associated BMF; if increased, consider iron deficiency or rare association with MPN
WBC	= to ↓	If reduced, consider associated BMF
Hemosiderinuria	↑ to ↑↑↑	Not routinely performed
Schistocytes	Absent	If present, consider alternative diagnosis (e.g., microangiopathies and intravascular devices)
Direct antiglobulin test	↓	If positive, consider AIHA; may be positive for C3d upon ECU treatment
Extravascular hemolysis	↑ to ↑↑	May be present, especially upon ECU treatment
Thrombosis	↑↑ (atypical sites)	Test for hereditary or acquired thrombophilia
Infections	↑↑	May be present, especially upon treatment with ECU or due to BMF; vaccines are indicated prior to ECU
Flow cYtometry	↑ to ↑↑↑	Clone size usually related with PNH subtype, anemia/hemolysis, and thrombotic risk

ECU: eculizumab; BMF: bone marrow failure syndrome; AIHA: autoimmune hemolytic anemia; MCV mean corpuscular volume; WBC white blood cells; ↑, ↑↑, ↑↑↑ indicate different levels of increase from normality; ↓, ↓↓, ↓↓↓ indicate different levels of decrease from normality; = indicates similarity with healthy status.

**Table 2 jcm-10-00948-t002:** Ongoing studies for new drugs for paroxysmal nocturnal hemoglobinuria.

Target of Inhibition	Drug	Company	Mechanism of Action	AD.	Phase of Study	Registered Number
c5	ABP959	Amgen (Thousand Oaks, CA, USA)	C5 Ab (Biosimilar)	IV	phase III	NCT03818607
c5	SB12	Samsung Bioepis	C5 Ab (Biosimilar)	IV	phase III	NCT04058158
C5	BCD-148	Biocad	C5 Ab (Biosimilar)	IV	phase III	NCT04060264
c5	Elizaria	Genirium	C5 Ab (Biosimilar)	IV	-	NCT04671810
c5	Ravulizumab, Ultomiris, ALXN1210	Alexion	C5 Ab (increased halflife)	IV and sc	Approved FDA; EMEA phase III for sc drug	NCT02946463 NCT03056040
c5	Crovalimab, SKY59, RO7112689	Roche	C5 Ab (increased halflife)	IV and sc	III	NCT04654468 NCT04432584 NCT04434092
c5	Tesidolumab, LFG316	Novartis	C5 Ab	IV	phase II	NCT02534909
c5	Pozelimab, REGN3918	Regeneron	C5 Ab	IV and sc	phase II	NCT03946748
c5	Zilucoplan, RA101495	Ra Pharma	C5 Small peptide	sc	phase II; oral formulation under development	NCT03225287
c5	Nomacopan, VA576, Coversin	Akari	C5 Small peptide	sc	phase III	NCT03829449
c5	Cemdisiran, ALN-CC5	Alnylam	C5RNAi	sc	phase I/II	NCT02352493
C3	Pegcetacoplan, APL-2	Apellis	Pegylated compstatin	sc	phase II/III	NCT04085601
Factor D	Danicopan, ACH-4471	Achillion/Alexion	Small peptide	oral	phase IIphase III	NCT03472885 NCT03181633 NCT04170023 NCT04469465
Factor D	BCX9930	Biocryst	Small peptide	oral	phase I	NCT04330534
Factor D	Iptacopan, LNP023	Novartis	Small peptide	oral	phase IIphase III	NCT03439839 NCT04558918

AD: way of administration; C5 Ab: anti-C5 monoclonal antibodies; IV: intravenous; sc: subcutaneous; C5RNAi: C5-RNA inhibitors; FD: factor D; FB: factor B; FDA: food and drug administration; EMEA European medical agency.

## Data Availability

Please refer to suggested Data Availability Statements in section “MDPI Research Data Policies” at https://www.mdpi.com/ethics (accessed on 22 January 2021).

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
