# Peer review of "Difficult Cases of Paroxysmal Nocturnal Hemoglobinuria: Diagnosis and Therapeutic Novelties"

_jcm, 2021, doi:10.3390/jcm10050948_

Round 1

Reviewer 1 Report

In this review authors describe old and new clinical and therapeutic aspects of PNH through a very interesting and practically useful  manner by presentation and discussion of instructive real clinical cases.

Major

The only important note concerns the importance of allogeneic bone marrow transplant, which  was almost completely ignored in the review. Although the authors found it worth reminding that the only curative treatment of PNH is allogeneic bone marrow transplant, they refer to outdated results with significant morbidity and mortality. It has been shown however in the recent years, that myeloablative conditioning regimen is not required to eradicate the PNH clone and reduced intensity conditioning regimens enabled to reduce transplant-related morbidity and mortality - reports have shown improved outcomes with survival rates in the 80 to 90 percent range. Patients who meet criteria for severe aplastic anemia with a PNH clone should be managed with either allogeneic HCT or immunosuppressive therapy for AA, and patients with a myelodysplastic syndrome and a PNH clone should receive appropriate supportive care or allogeneic HCT as dictated by their MDS.

So it should be clear from the article, that transplant option should be also considered if aplastic anemia or myelodysplastic syndrome is diagnosed.

Minor

Error in line 78: the assumption that (not “than”)

Author Response

Referee 1

In this review authors describe old and new clinical and therapeutic aspects of PNH through a very interesting and practically useful manner by presentation and discussion of instructive real clinical cases.

Major

The only important note concerns the importance of allogeneic bone marrow transplant, which was almost completely ignored in the review. Although the authors found it worth reminding that the only curative treatment of PNH is allogeneic bone marrow transplant, they refer to outdated results with significant morbidity and mortality. It has been shown however in the recent years, that myeloablative conditioning regimen is not required to eradicate the PNH clone and reduced intensity conditioning regimens enabled to reduce transplant-related morbidity and mortality - reports have shown improved outcomes with survival rates in the 80 to 90 percent range. Patients who meet criteria for severe aplastic anemia with a PNH clone should be managed with either allogeneic HCT or immunosuppressive therapy for AA, and patients with a myelodysplastic syndrome and a PNH clone should receive appropriate supportive care or allogeneic HCT as dictated by their MDS.

So it should be clear from the article, that transplant option should be also considered if aplastic anemia or myelodysplastic syndrome is diagnosed.

We thank the Referee for the revision and for the important suggestions. We fully agree that it is imperative to better describe the utility of bone marrow transplant in patients with BMF and concomitant PNH. The paragraph was rephrased as follows:

“Finally, it is worth reminding that the only curative treatment of PNH is allogeneic bone marrow transplant (BMT). BMT outcome significantly improved after the introduction of reduced intensity conditioning and the addition of ECU reduced the morbidity and mortality linked to PNH [Brodsky Blood 2021]. On the whole, BMT is indicated in patients with PNH and severe aplastic anemia below 40 years of age and with available donor and has the potential to eradicate both aplastic anemia and the PNH clone. Also, PNH-MDS patients may benefit from BMT if indicated for MDS itself (i.e. high-risk patients or low-risk MDS with life-threatening cytopenias and/or molecular abnormalities.”

Brodsky RA. How I Treat Paroxysmal nocturnal hemoglobinuria. Blood. 2021 Jan 21:blood.2019003812. doi: 10.1182/blood.2019003812. Epub ahead of print. PMID: 33512400.

Minor

Error in line 78: the assumption that (not “than”)

Thank you. The mistake has been corrected.

Reviewer 2 Report

This is a nice up-to-date overview of PNH with some thoughtful clinical vigniettes. I've edited the manuscipt a bit and added some suggestions as depiceted in the attached PDF file

Author Response

Referee 2

This is a nice up-to-date overview of PNH with some thoughtful clinical vigniettes. I've edited the manuscipt a bit and added some suggestions as depiceted in the attached PDF file

We are extremely grateful to the Referee for the careful and thorough proof-reading and for the important suggestions. We revised the manuscript accordingly, including the addition of the suggested references.

Not true in all cases! oral FD and FB inhibitors have to be given twice daily! this paragraph has to be rephrased

We thank the Referee and rephrased the paragraph.

“Preclinical and clinical studies have demonstrated that novel C5 inhibitors (ECU biosimilars, ECU with longer half-life such as ravulizumab iv or sc [22], small molecules [23, 24] and siRNA [25]) effectively inhibit complement mediated-hemolysis and may imply more manageable administration routes/schedules; in particular, ravulizumab has a longer half-life, and was shown to be not inferior to ECU and to reduce the incidence of BTH. Additionally, some of the new agents also proved effective in patients with C5 polymorphisms [20].”

This is true, however, the only results published so far are the two RAVU trials, In addtion, new data have just identified C3 "bypass activation" of C5 which might be one of the reasons why there are some patients under C3 Inhibtion still experience BTH.

We thank the Referee for the important hint and added a sentence about the bypass activation of C5 with the relative citations.

“However, residual C5 activation may still occur despite upstream complement inhibition and Mannes et al recently identified a “C3 bypass” model of terminal complement activation. This happens because of a conformational change in C5, which adopts a C5b-like structure on highly opsonized surfaces allowing the formation of MAC complex. These findings may explain residual complement activation upon both C5, C3 and C1 inhibitors [Mannes et al Blood 2021, Roumenina Blood 2021].”

Roumenina LT. Terminal complement without C5 convertase? Blood. 2021 Jan 28;137(4):431-432. doi: 10.1182/blood.2020010133. PMID: 33507301.

Mannes M, Dopler A, Zolk O, Lang SJ, Halbgebauer R, Höchsmann B, Skerra A, Braun CK, Huber-Lang M, Schrezenmeier H, Schmidt CQ. Complement inhibition at the level of C3 or C5: mechanistic reasons for ongoing terminal pathway activity. Blood. 2021 Jan 28;137(4):443-455. doi: 10.1182/blood.2020005959. PMID: 33507296.

Reviewer 3 Report

The authors reported a review on difficult cases of paroxysmal nocturnal hemoglobinuria taking account especially diagnosis and therapeutic novelties.

The review was well written and easy to follow. It described well pathogenesis. Clinical vignettes were informative and they were described in the logical order. This review helps reader to do clinical decision. I really enjoyed reading this review. Congratulations for the nice article!

Author Response

We thank the Referee for the positive comments and are extremely pleased that the article was enjoyable.

Round 2

Reviewer 1 Report

The authors fully responded to the comments contained in the review